# Sterols and Triterpenes from *Dobera glabra* Growing in Saudi Arabia and Their Cytotoxic Activity

**DOI:** 10.3390/plants10010119

**Published:** 2021-01-08

**Authors:** Wael M. Abdel-Mageed, Ali A. El-Gamal, Shaza M. Al-Massarani, Omer A. Basudan, Farid A. Badria, Maged S. Abdel-Kader, Adnan J. Al-Rehaily, Hanan Y. Aati

**Affiliations:** 1Department of Pharmacognosy, College of Pharmacy, King Saud University, P.O. Box 2457, Riyadh 11451, Saudi Arabia; wabdelmageed@ksu.edu.sa (W.M.A.-M.); salmassarani@ksu.edu.sa (S.M.A.-M.); basudan@ksu.edu.sa (O.A.B.); ajhmkl@hotmail.com (A.J.A.-R.); hati@ksu.edu.sa (H.Y.A.); 2Pharmacognosy Department, Faculty of Pharmacy, Assiut University, Assiut 71526, Egypt; 3Department of Pharmacognosy, Faculty of Pharmacy, Mansoura University, El-Mansoura 35516, Egypt; faridbadria@gmail.com; 4Pharmacognosy Department, College of Pharmacy, Sattam Bin Abdulaziz University, Al-kharj 11942, Saudi Arabia; mpharm101@hotmail.com; 5Department of Pharmacognosy, College of Pharmacy, Alexandria University, Alexandria 21215, Egypt

**Keywords:** *Dobera glabra*, salvadoraceae, triterpenes, steroids, cytotoxic activity, phosphodiesterase inhibition

## Abstract

A new lupane caffeoyl ester, lup-20(29)-ene 3β-caffeate-30-al (**7**), and a new oleanane-type triterpene, 3*β*-hydroxyolean-13(18)-en-12-one (**17**), were isolated from the aerial parts of *Dobera glabra* (Forssk), along with ten known triterpenes, including seven lupane-type lupeol (**1**), 30-nor-lup-3*β*-ol-20-one (**2**), ∆^1^-lupenone (**3**), lup-20(29)-en-3*β*,30-diol (**4**), lupeol caffeate (**5**), 30-hydroxy lup-20(29)-ene 3*β*-caffeate (**6**), and betunaldehyde (**8**); three oleanane-type compounds were also identified, comprising *δ*-amyrone (**15**), *δ*-amyrin (**16**), and 11-oxo-*β*-amyrin (**18**); together with six sterols, comprising *β*-sitosterol (**9**), stigmasterol (**10**), 7*α*-hydroxy-*β*-sitosterol (**11**), 7*α*-hydroxy-stigmasterol (**12**), 7-keto-*β*-sitosterol (**13**), and 7-keto-stigmasterol (**14**). Their structures were elucidated using a variety of spectroscopic techniques, including 1D (^1^H, ^13^C, and DEPT-135 ^13^C) and 2D (^1^H–^1^H COSY, ^1^H–^13^C HSQC, and ^1^H–^13^C HMBC) nuclear magnetic resonance (NMR) and accurate mass spectroscopy. Subsequently, the different plant extracts and some of the isolated compounds (**1–9**, **11** and **13**) were investigated for their possible cytotoxic activity in comparison to cisplatin against a wide array of aggressive cancer cell lines, such as colorectal cancer (HCT-116), hepatocellular carcinoma (HepG-2), and prostate cancer (PC-3) cell lines. Compound **11** displayed broad cytotoxicity against all of the tested cell lines (IC_50_ ≅ 8 µg/mL in all cases), and a high safety margin against normal *Vero* cells (IC_50_ = 70 µg/mL), suggesting that **11** may be a highly selective and effective anticancer agent candidate. Notably, the evidence indicated that the mode of action of compound **11** could possibly consist of the inhibition of phosphodiesterase I (80.2% enzyme inhibition observed at 2 µM compound concentration).

## 1. Introduction

The Salvadoraceae plant family comprises three genera—*Azima, Dobera*, and *Salvadora*—with around twelve species, distributed in the hot and dry areas of mostly mainland Africa, Madagascar, Southeast Asia, the Indonesian island of Java, and Malaysia [1,2]. The Salvadoraceae family is represented in Saudi Arabia by two genera: *Dobera* and *Salvadora;* both are dominant near the foothills, where Wadi Jizan originate.

*Dobera* is a small genus comprising only two species, *Dobera loranthifolia* and *Dobera glabra*, which are endemic to East Africa and North West India. *D. glabra* (Forssk) is native to many African countries, such as Djibouti, Ethiopia, Somalia, and Sudan, but it is also found in India and Saudi Arabia. It is a highly valued plant, and it is the only *Dobera* species to be found in Saudi Arabia, usually in alluvial areas, on slopes, or in Wadis like Wadi Tashar and Wadi Kawbah, in the regions near the border with Yemen [3]. This plant species might be endangered, as the local people declared that there is no new generation of the trees, and only the old trees are sparingly distributed; in a comparatively recent study, the survival rate of seedlings and samplings was observed to be greatly decreasing, and the rate of fruit production was observed to be minimal [4]. *D. glabra*, known as Dobar in Arabic, is a fair-sized evergreen shrub or tree with thick, leathery, opposite leaves, sweet-scented white flowers, and purple ovate fruits with a layer of jelly-like edible fluid around a single flat seed. The fruits are characterized by a bad smell, and they are considered, together with the seeds, typical famine food consumed during times of drought in Ethiopia and many other African countries [5,6]. However, excessive ingestion causes stomach aches and intestinal problems [7]. Notably, the tree is highly valued in folkloric medicine, since its latex is applied to both eyes once daily for three days for the treatment of ophthalmic problems [8], whereas the plant’s flowers provide an essential oil used as perfume [9]. In Somalia, it is mainly used as shade for the farmer and his livestock. Its bush fallow is often used to maintain soil fertility [10], and in Kenya, it is used as a diet and fodder tree [11].

Despite the importance of *D. glabra* as an edible plant for humans and animals, our literature search revealed only one phytochemical study conducted on this plant; it reported the isolation of seven flavonoids from the leaves of *D. glabra*, which displayed antioxidant activity and genotoxic protection against CCl_4_-induced liver damage in male rats [12].

Cancer is a major public health problem worldwide and is the second leading cause of death in the United States. As part of our intensive search for new bioactive compounds in Saudi plants with a potential cytotoxic activity, in the current study we thoroughly investigated *D. glabra* grown in the wild in Saudi Arabia in order to identify its active constituents and assess their cytotoxic activity. All of the obtained fractions, as well as some of the isolated compounds chosen based on their availability and the reported activity of structurally-related steroids and triterpenes, were screened for cytotoxic activity [13,14]. The biologically-guided fractionations of the different active fractions were subjected to further chromatographic isolation and separation.

## 2. Results and Discussion

### 2.1. Structure Elucidation of the New Triterpenoids

The investigation of the phytochemicals isolated from the aerial parts of *D. glabra* via a combination of different chromatographic methods led to the isolation of 18 triterpene and steroid compounds (**1**–**18**) (Figure 1). Two of the isolated compounds (**7** and **17**) are hereby reported for the first time as being isolated from a natural source, whereas the remaining compounds are hereby reported for the first time as being isolated from *D. glabra* [12]. The structures of the isolated compounds were elucidated via the extensive use of various spectroscopic techniques, including 1D (^1^H, ^13^C, and DEPT-135 ^13^C) and 2D (^1^H–^1^H COSY, ^1^H–^13^C HSQC, and ^1^H–^13^C HMBC) nuclear magnetic resonance (NMR) (Appendix A) and accurate mass measurements, as well as by comparing the compounds’ physical and spectral characteristics with those reported in the literature for previously-isolated compounds. The known compounds were identified as lupeol (**1**) [15,16,17], 30-norlup-3β-ol-20-one (**2**) [18], ∆^1^-lupenone (glochidone) (**3**) [19], lup-20(29)-ene-3*β*,30-diol (29-hydroxylupeol) (**4**) [20], lupeol caffeate (**5**) [17,21], 30-hydroxy lup-20(29)-ene 3*β*-caffeate (**6**) [22] and betunaldehyde (**8**) [23]; three oleanane-type compounds were also identified: *δ*-amyrone (**15**) [24], *δ*-amyrin (**16**) [24], and 11-oxo-*β*-amyrin (**18**) [25]; in addition, six sterols were identified: *β*-sitosterol (**9**) [26], stigmasterol (**10**) [27], 7*α*-hydroxy-*β*-sitosterol (**11**) [28], 7*α*-hydroxy-stigmasterol (**12**) [29], 7-keto-*β*-sitosterol (**13**) [30,31], and 7-keto-stigmasterol (**14**) [32]

Compound **7** was obtained as a pale yellow amorphous solid. Based on the high-resolution electron ionization mass spectrometry (HREIMS) evidence, the molecular formula of this compound was determined to be C_39_H_54_O_5_, derived from the quasi-molecular ion peak (m/z 603.4048 [M + H]^+^), implying thirteen degrees of unsaturation. The infra-red (IR) spectrum of **7** included absorption bands at 3481, 1705, and 1688 cm^−1^, which were assigned to the OH, conjugated CO ester, and aldehydic C = O groups, respectively; the ultraviolet (UV) spectrum of **7** included absorption bands at λ_max_ = 224, 246, 298, and 330 nm.

The ^13^C NMR spectrum of compound **7** (Table 1), with the aid of the Distortionless Enhancement of Polarization Transfer using a 135 degree decoupler pulse (DEPT-135) and ^1^H–^13^C heteronuclear single quantum coherence (HSQC) experiments, was comprised of the resonance signals of 39 carbons, which were identified as six methyls, 11 methylenes (ten aliphatic and one vinylic), 12 methines (five aliphatic, one O-bearing at *δ_C_* 81.5, two olefinic at *δ_C_* 115.8 and 145.2, three aromatic at *δ_C_* 114.4, 115.5 and 122.4, and one aldehydic at *δ_C_* 196.0), and 10 quaternary carbon atoms (five aliphatic carbons, one vinylic carbon at *δ_C_* 157.5, three aromatic carbons at *δ_C_* 127.3, 144.4, and 146.9, and one carbonyl carbon at *δ_C_* 168.3). Moreover, the ^1^H NMR spectrum (Table 1) of **7** comprised one characteristic aldehyde proton signal at *δ_H_* 9.65 (s), two deshielded exocyclic methylene protons at *δ_H_* 6.48 and 6.11 (1H each, brs), one O-bearing methine proton at *δ_H_* 4.72 (t, *J* = 8.1), and six tertiary methyl group protons at *δ_H_* 0.96 (s), 0.99 (s), 1.01 (s), 1.04 (s), 1.06 (s), and 1.14 (s). In addition, two H-atom signals associated to a *trans* olefinic bond were observed at *δ_H_* 6.39 (d, *J* = 15.5) and 7.70 (d, *J* = 15.5); these signals were accompanied by three aromatic signals assignable to a 1,3,4-trisubstitued phenyl ring at *δ_H_* 7.02 (d, *J* = 7.5), 7.11 (d, J = 7.5), and 7.26 (brs), indicating the presence of a caffeoyl moiety [17]. The aforementioned data suggest that **7** is a pentacyclic triterpenoid caffeate.

The assignments of the signals due to the methyl groups and the remaining proton and carbon signals were performed through ^1^H–^1^H COSY and ^1^H–^13^C HMBC experiments. Ultimately, these data enabled us to conclude the gross structure of **7** to be that of a lupane-type triterpenoid caffeate (Figure 2) [17,21,22]. Key ^1^H–^13^C HMBC cross-peaks were observed between H_3_-23, H_3_-24 (*δ_H_* 1.01, 1.04, respectively) and C-3, C-5; H_3_-25 (*δ_H_* 0.99) and C-5, C-9; H_3_-26 (*δ_H_* 1.14) and C-9; H_3_-27 (*δ_H_* 1.06) and C-8, C-13, C-14, and C-15; H_3_-28 (*δ_H_* 0.96) and C-16, C-17, and C-22; and H_2_-29/H-30 and C-20. Based on the above-detailed observations, the planar structure of the triterpenoid moiety of **7** was identified to be 3β-hydroxylup-20(29)-en-30-al. The caffeoyl unit is proposed to be linked to the C-3 of the triterpenoid moiety, given that the NMR resonance of H-3 was downfield-shifted to *δ_H_* 4.72 (t, *J* = 8.1) in **7** with respect to compound **1** (*δ_H_* 3.16 (dd, *J* = 8.1)), and the resonance of C-3 was downfield-shifted to *δ_C_* 81.5 in **7** compared to *δ_C_* 79.04 in **1**. This conclusion was ultimately confirmed by the observed HMBC correlation between H-3 and C-1’ (*δ_C_* 168.3). Furthermore, the aldehyde group proton (*δ_H_* 9.65 (s)) was assigned at C-30, as inferred via the ^2^*J*_CH_ HMBC correlation between H-30 and C-20 (*δ_C_* 157.5), as well as the ^3^*J*_CH_ HMBC correlation between vinylic protons H_2_-29 and C-30. Accordingly, **7** was elucidated to be lup-20(29)-en-30-al 3*β*-caffeate.

Compound **17** was obtained as a white amorphous solid. Its molecular formula was determined to be C_30_H_48_O_2_, based on the HR-ESI-MS data (*m/z* 441.3733 ([M + H]^+^; calc. 441.3733), implying seven degrees of unsaturation. The IR spectrum of **17** included absorption bands at 3436 and 1663 cm^−1^, which were assigned to a hydroxyl group and a conjugated ketone, respectively.

The ^1^H NMR spectrum of **1****7** included resonance signals due to one oxygenated methine at *δ*_H_ 3.16 (1H, dd, *J*  =  11.0, 4.8 Hz, H-3), and eight methyl protons at *δ*_H_ 0.79 (3H, s, H-24), 0.87 (3H, s, H-29), 0.93 (3H, s, H-30), 0.95 (3H, s, H-25), 0.97 (3H, s, H-27), 0.99 (3H, s, H-23), 1.06 (3H, s, H-26), and 1.11 (3H, s, H-28). The ^13^C NMR spectrum of **1****7** was comprised of resonance signals due to 30 carbons, including a conjugated ketone carbonyl group at *δ*_C_ 209.9 (C-12), two olefinic carbons at *δ_C_* 140.6 and 149.6 of four substituted double bonds, eight methyls at *δ*_C_ 32.7 (C-30), 28.6 (C-23), 24.9 (C-29), 23.7 (C-28), 22.1 (C-27), 17.5 (C-26), 16.4 (C-25), and 16.2 (C-24), three methines (two aliphatic and one O-bearing at *δ_C_* 79.4), ten methylenes, and six aliphatic quaternary carbons (Table 1). The above-mentioned spectroscopic data indicate that compound **17** is a pentacyclic triterpene.

The assignment of the signals of the methyl groups and the remaining of proton and carbon signals was performed through ^1^H–^1^H COSY and ^1^H–^13^C HMBC experiments; the data collected in this way indicated the gross structure of **17** to be that of an oleane-type triterpenoid [16] (Figure 2). Key HMBC correlations were observed between H_3_-23, H_3_-24 (*δ_H_* 0.99, 0.79, respectively) and C-3 (*δ_C_* 79.4), C-5 (*δ_C_* 56.7); H_3_-25 (*δ_H_* 0.95) and C-1 (*δ_C_* 39.5), C-5 (*δ_C_* 56.7), and C-9 (*δ_C_* 51.6); H_3_-26 (*δ_H_* 1.06) and C-8 (*δ_C_* 42.0), C-9 (*δ_C_* 51.6), and C-14 (*δ_C_* 46.6); H_3_-27 (*δ_H_* 0.97) and C-8 (*δ_C_* 42.0), C-13 (*δ_C_* 140.6), C-14 (*δ_C_* 46.6), and C-15 (*δ_C_* 26.2); H_3_-28 (*δ_H_* 1.11) and C-17 (*δ_C_* 36.2), C-18 (*δ_C_* 149.6), and C-22 (*δ_C_* 40.1); H_3_-29 (*δ_H_* 0.87)/H_3_-30 (*δ_H_* 0.93) and C-19 (*δ_C_* 40.5), C-20 (*δ_C_* 34.7), and C-21 (*δ_C_* 36.4). Furthermore, a key ^2^*J*_CH_ HMBC correlation between H_2_-11 (*δ_H_* 2.30) and C-9/C-12 (*δ_C_* 209.9), accompanied by a ^2^*J*_CH_ HMBC correlation from H_2_-19 (*δ_H_* 1.67) to C-18, and a ^3^*J*_CH_ HMBC correlation from H_2_-19/H_3_-27 and C-13, and from H_3_-28 to C-18 indicated the presence of a 12-oxo-olean 13(18)-ene system [24,33]. Based on the above observations, the structure of **17** was deduced to be that of 12-oxo-olean-13(18)-en-3*β*-ol (12-oxo-*δ*-amyrin).

### 2.2. Cytotoxic Activity

The antitumor activities of the different fractions and isolated compounds were tested against different cancer cell lines (HCT-116, PC-3, and HepG-2). The data indicated that compound **11** showed broad-spectrum activity on three different cell lines. Interestingly, all of the extracts exhibited good (CHCl_3_ extract) to moderate (n-hexane, butanol, and ethanol extracts) activity. On the other hand, most of the pure compounds (**1****–4**, **8**, and **9**) displayed no activity, whereas compounds **5****–7** and **13** displayed moderate activity (Table 2). Perhaps the higher activity of all of the extracts against the pure compounds was due to synergistic effect of all of the secondary metabolites in the whole extracts, rather than one compound. According to the obtained results (Table 2), the sterol nucleus, in general, exhibited stronger activity than that of lupane triterpene. For the sterol, the maximum activity was observed with the 7α-enol system. The oxidation of the 7α-hydroxy group to form the 7-enone system reduced the activity, while the activity of the lupane nucleus is improved by acylation with caffeic acid at OH-3 [13,14].

In order to investigate the selectivity of the most active compounds for cancer cell lines, and to demonstrate that they had no cytotoxic effects on normal (non-cancerous) cells, viability and wound-healing assays were performed. Over 95% cell viability (in *Vero* cells) was obtained at a 70 µM concentration of the most active compounds, so the biocompatibility was confirmed by conducting the wound-healing assay at either 35 or 70 µM concentrations of each compound. The cells treated with compound **11** were able to heal the wound at a rate closer to that observed for untreated cells (Figure 3). These data indicated that **11**, the most active compound, is not cytotoxic to normal cells, and that it is worth investigating further for use as a safe anticancer agent.

### 2.3. Possible Cytotoxicity Mechanism of Compound ***11***

The bioactivity of compound **11** was further investigated using phosphodiesterase, and this compound displayed an 80.2% inhibitory activity toward the mentioned enzyme in the used condition. Although high intracellular levels of cAMP have been reported to effectively inhibit the proliferation of cancer cells, compounds that cause cAMP levels to be elevated are not recommended as anticancer drugs, due to their high cytotoxicity [34,35,36].

All of the known phosphodiesterase inhibitors operate via three main types of interactions: interactions with metal ions mediated through water, H-bond interactions with protein residues involved in nucleotide recognition, and, most importantly, interactions with the enzyme’s hydrophobic residues; therefore, these interactions should guide the design of new classes of inhibitors [37].

Our research team, which consists of a multidisciplinary collection of international scientists, is developing more selective and effective anticancer agents than those that are available today, spurred on by the increasing need for safe and efficacious agents for cancer therapy. In this context, compound **11** proved to be an efficient and selective agent that is worth considering for further in vivo studies as a cancer treatment.

#### Phosphodiestrase Inhibition Investigation

Compound **11**, which proved to be the most active and selective anticancer agent among all of the compounds tested in the present study, showed a remarkable inhibitory activity against phosphodiesterase I (PDE1), with an 80.2% inhibition of this enzyme at a 2 µM concentration of the compound.

The in vitro assay revealed **11** to be a highly selective anticancer agent, with a noticeable cytotoxic activity against colorectal cancer (HCT-116), hepatocellular carcinoma (HepG-2), and prostate cancer (PC-3) cells compared to the commonly-used chemotherapeutic cisplatin drug.

Additionally, in the PDE1 inhibition tests, compound **11** exhibited a high cytotoxic activity against colorectal, prostate, liver cancers, with high selectivity. This high selectivity and promising activity against three aggressive cancer cell lines renders compound **11** a promising anticancer agent candidate. Therefore, the use of this compound in combination with other chemotherapeutic drugs should be carefully investigated as a way to explore the possibility of developing chemotherapeutic cancer treatments with increased efficacy and reduced undesired side-effects.

## 3. Materials and Methods

### 3.1. Instrumentation and Chemicals

The Fourier-transform infrared (FT-IR) spectra were recorded on a Nicolet 5700 FT-IR Microscope spectrometer (FT-IR Microscope Transmission, company, Waltham, MA, USA). The optical rotations were measured using a Perkin-Elmer Model 341 LC polarimeter (PerkinElmer, MA, USA). The accurate mass determination was achieved with a JEOL JMS-700 High-Resolution Mass Spectrophotometer (JEOL USA Inc., Peabody, MA, USA) with a positive and negative mode. The NMR spectroscopy experiments were carried out using deuterated chloroform and an UltraShield Plus 500 (Bruker) spectrometer operating at 500 MHz for ^1^H and 125 MHz for ^13^C at the College of Pharmacy, Sattam Bin Abdulaziz University. The microplate reader used was a Coming Inc., NY, USA, ELISA BioTek L × 800 microplate. The thin layer chromatography (TLC) was performed on normal and reversed phase silica gel (Merck, Darmstadt, Germany) with a layer thickness of 250 μm and a mean particle size of 10–12 μm, with different dimentions. *n*-hexane:EtOAc, CHCl_3_:EtOAc and CHCl_3_:MeOH were used for the normal phase TLC, while H_2_O: MeOH (10:90) was used for reversed phase RP-18 TLC. Additionally, the compounds were visualized by spraying the TLC plates with 15% H_2_SO_4_/ethanol, or with anisaldehyde-sulfuric acid, followed by heating. The column chromatography was carried out on silica gel (Merck 60 A, 230–400 mesh ASTM, Darmstadt, Germany), while LiChrorep RP-18 (25–40 µm) was used for the reversed phase column chromatography.

The centrifugal preparative TLC (CPTLC) was performed on a chromatotron instrument (Harrison Research, Palo Alto, California, CA, USA). Plates coated with 1 and 2 mm of silica gel 60, 0.04–0.06 mm were used. The reagents, chemicals, and solvents were of analytical grade, and they were purchased from Sigma-Aldrich, Loba Chemie Pvt. Ltd., and SD Fine Chem. Ltd. The water was doubly distilled before use [38,39].

### 3.2. Plant Material

The aerial parts of *D. glabra* were collected from the Shoqaiq in February 2013. The specimen of the plant was identified by Mohamed Yousef, Professor of Taxonomy at the Department of Pharmacognosy, College of Pharmacy, King Saud University. A voucher specimen No. 16,036 of *D. glabra* was deposited at the herbarium of the Pharmacognosy Department, College of Pharmacy, King Saud University, Kingdom of Saudi Arabia. The undesirable parts of plant material were removed. The aerial parts were dried in air-shade until they reached a constant weight, after which they were ground using a toothed mill, followed by sifting using suitable mesh in order to give a homogenous particle size powder for the subsequent efficient extraction.

### 3.3. Extraction and Isolation

The air-dried and powdered aerial parts of *D. glabra* (1200 g) were extracted by maceration with 96% ethanol. The extract thus obtained was evaporated in vacuo to yield a brownish residue (28 g), which was suspended in water and subsequently partitioned with *n*-hexane (3 g), chloroform (4.5 g), and *n*-butanol (0.5 g), in succession.

The *n*-hexane fraction was purified by chromatography over a silica gel column and eluted with CHCl_3_:EtOAc mixtures of increasing polarity. On the basis of the TLC behavior, the appropriate fractions were combined in order to give seven main fractions (H_1–_H_7_). The columns were monitored by examination under a UV lamp 254/366 nm, followed by spraying with 15% H_2_SO_4_ in ethanol, followed by heating at 120 °C. Fractions H_1_–H_3_ were eluted with CHCl_3_:EtOAc (98:2–95:5), and yielded compounds **1** (50.0 mg), **9** (54.0 mg), and **10** (40.0 mg), after solvent treatment. Fraction H_4_ was purified by chromatography over an RP-18 column using CH_3_CN only as the eluent to afford compounds **2** (11.5 mg) and **3** (9.5 mg). Fraction H_5_ was eluted with CHCl_3_:EtOAc (98:2), and was further purified using a silica gel RP-18 column using methanol (MeOH) as the eluent to afford compounds **15** (33.6 mg) and **16** (18.7 mg). Fraction H_6_ was subjected to a chromatotron (CPTL, silica gel 60 GF_254_, 1mm, solvent: CHCl_3_:EtOAc (97:3)); this procedure yielded 31.0 mg of compound **13** and 12.7 mg of compound **14**. Fraction H_7_ was eluted with CHCl_3_:EtOAc (90:10), and was subjected to chromatography over a silica gel RP-18 column using MeOH only as the eluent, to yield compounds **11** (3.0 mg) and **12** (1.6 mg).

The CHCl_3_ fraction was purified by chromatography over a silica gel column, and was eluted with CHCl_3_: MeOH mixtures of increasing polarity in order to yield five main fractions (C_1_–C_5_). The crystallization of C_1_ in CHCl_3_:MeOH yielded compound **4** (13.0 mg); on the other hand, the filtrate of this fraction was subjected to CPTLC (silica gel 60 GF_254_, 1 mm, solvent: hexane:EtOAc (80:10)) to yield 5 mg of compound **18**. Fraction C_2_ was purified by CPTLC (silica gel 60 GF_254_, 1 mm, solvent: hexane:EtOAc (70:30)) to give compound **8** (8.7 mg).

Fractions C_3_ and C_4_ were re-chromatographed separately over a silica gel RP-18 column, using MeOH only as the eluent, to yield compounds **5** (14.0 mg), **6** (10.1 mg), and **7** (10.5 mg). The elution of the silica gel RP-18 column of fraction C_5_ with H_2_O: MeOH (10:90) afforded compound **17** (5.7 mg).

### 3.4. Cytotoxic Activity

#### 3.4.1. Cell Lines and Tested Compounds

The cytotoxic activity of the different fractions and the isolated compounds was tested against different human cancer cells—that is, prostate carcinoma cells (PC-3), hepatocellular carcinoma cells (HepG-2), and colorectal cancer cell (HCT-116)—as well as against the African green monkey kidney cell line (Vero-B). The cell lines were obtained from the American Type Culture Collection. The cells were cultivated at 37 °C and 10% CO_2_ in Dulbecco’s Modified Eagle Medium (Lonza, 12-604F) supplemented with 10% fetal bovine serum (Lonza, Cat. No.14-801E), 100 IU/mL pencillin and 100 µg/mL streptomycin (Lonza, 17-602E). Cisplatin (*cis*-diamineplatinum (II) dichloride), obtained from sigma, was dissolved in 0.9% saline, then stored as an 8 mM stock solution at −20 °C and used as the positive control.

The tested compounds were solubilized in dimethyl sulfoxide (DMSO) and stored at −20 °C. A 0.5% solution of crystal violet was prepared in MeOH and used to stain the viable cells [40,41,42,43]. Notably, crystal violet binds to proteins and DNA in adherent and viable cells, so this staining is indicative of the viability of the treated cells. The viability of the cells was quantified using the MTT reagent, which contains 3-(4,5-dimethylthiazol-2-yl)-2,5-diphenyl tetrazolium bromide, and measures the activity of mitochondrial dehydrogenase in viable cells.

#### 3.4.2. Cell Cultures

The cells were seeded in a 96-well plate as 5 × 10^4^ cells/mL (100 µL/well). In total, 100 µL/well of from the serial dilutions of the tested compounds and cisplatin (100, 30, 10, 3.3, 1.1, or 0.37 µM) were added to the plate after the overnight incubation of the cells at 37 °C and 5% CO_2_. DMSO was used as a control (0.1%). The cells were incubated for 48 h. Subsequently, 15 µl MTT (5 mg/mL PBS, phosphate buffered saline) was added to each well, and the plate was incubated for another 4 h. The formazan crystals were solubilized in 100 µL acidified sodium dodecyl sulfate (SDS) solution (10% SDS/0.01 M HCl). After 14 h of incubation at 37 °C and 5% CO_2_, the absorbance of the wells was measured at 570 nm using a Biotech plate reader. Each experiment was repeated three times, and the standard deviation was calculated (±). The concentration that caused a 50% inhibition of the cell growth (IC_50_) was calculated for each compound or fraction. The growth of the cells was monitored and the images were acquired using Gx microscopes (GXMGXD202 Inverted Microscope) (10x Eyepiece) after staining with crystal violet [44].

### 3.5. Phosphodiestrase Inhibition Investigation

The phosphodiesterase I inhibition assay was performed using snake venom according to a previously-reported method, with minute variations. Briefly, Tris–HC1 buffer 33 mM at pH 8.8 (97 μL), 30 mM Mg acetate with an enzyme concentration of 0.742 µU well-1, and 0.33 mM bis-(*p*-nitrophenyl) phosphate (Sigma N-3002, 60 μL) as the substrate were taken. An EDTA solution characterized by an IC_50_ ± SD value of 274 ± 0.007 µM was used as the positive control. After a pre-incubation period of 30 min, the enzyme with the test samples was observed spectrophotometrically in order to detect its enzymatic activity on a microtitre plate reader at 37 °C. In particular, the rate at which the optical density of the sample changed (in min^−1^) was followed at 410 nm, which is a wavelength absorbed by the *p*-nitrophenol released from *p*-nitrophenyl phosphate, a reaction known to be catalyzed by phosphodiesterase I. All of the assays were processed in triplicate [34,35,36,37].

### 3.6. Wound-Healing Assay

WI-38 cells were seeded in a 6-well plate at 20 × 10^4^ cells per ml (2 mL in each well), which was incubated overnight at 37 °C and 5% CO_2_. During the second day, a scratch was created in each well with a p200 tip; the medium was then replaced with fresh medium containing either DMSO or different concentrations of the most active compounds. Images were recorded at different time points (0, 4, 24, and 48 h) in order to monitor the wound closure. Subsequently, the cells were washed twice with ice-cold 1X PBS and fixed with ice-cold MeOH for 20 min at 4 °C. The fixed cells were washed twice with 1X PBS and stained with 0.5% crystal violet for 30 min. Any unreacted crystal violet was washed off with distilled H_2_O until no color was observed in the washing. The size of the wound was measured using Image J1.47 software [45].

### 3.7. Analytical Data for Compound ***7***

Compoud 7 is a pale yellow amorphous solid (10.5 mg); [α]^23^_D_ +17.4° (*c* 0.5, MeOH); UV λ_max_ MeOH nm (log *ε*): 224 (4.13), 246 (3.76), 298 (3.55), and 330 (3.82). IR (KBr) *v*_max_ 3481, 2938, 1705, 1688, 1617, 1522, 1447, 1385, 1267, 1183, 871 cm^−1^; ^1^H and ^13^C NMR (see Table 1 and Table 2); HR-ESI-MS [M + H]^+^
*m/z* 603.4048 (calculated for C_39_H_54_O_5_, 603.4049).

### 3.8. Analytical Data for Compound ***17***

Compound 17 is a white amorphous solid (5.7 mg); [α]^23^_D_ +31° (*c* 0.8, CHCl_3_); UV λ_max_ MeOH nm (log *ε*): 253 (3.84). IR (KBr) *v*_max_ 3436, 2941, 2865, 1663, 1610, 1460, 1385, 1253, 1177 cm^−1^; ^1^H and ^13^C NMR (see Table 1 and Table 2); HR-ESI-MS [M + H]^+^
*m/z* 4413.3733 (calculated for C_30_H_48_O_2_, 441.3733).

## 4. Conclusions

Two compounds (**7** and **17**) were isolated for the first time from the aerial parts of *D. glabra,* and from a natural source. Additionally, a series of cytotoxic triterpenes and sterols, which were not previously known to be found in *D. glabra*, were also isolated from the mentioned plant parts. The structures of the new triterpenes were determined using a range of spectroscopic techniques, including high-resolution mass spectrometry. The different plant extracts and some of the isolated compounds were tested for their cytotoxic activity against colorectal cancer (HCT-116), hepatocellular carcinoma (HepG-2), and prostate cancer (PC-3) cell lines. The cytotoxic potency and selectivity of the bioactive compounds were investigated by the phosphodiestrase enzyme inhibition method. Compound **11** displayed a remarkable inhibitory activity against PDE1. The use of this compound in combination with other chemotherapeutic drugs should thus be investigated, in order to explore the possibility of producing chemotherapeutic cancer treatments characterized by increased efficacy and reduced undesired side effects.

Currently, a more rigorous in vivo study is underway, which is directed at obtaining more preclinical information, such as oral stability, bioavailability, and pharmacokinetic data, with the anticipation of better activity and a high safety margin.

## Figures and Tables

**Figure 1 plants-10-00119-f001:**
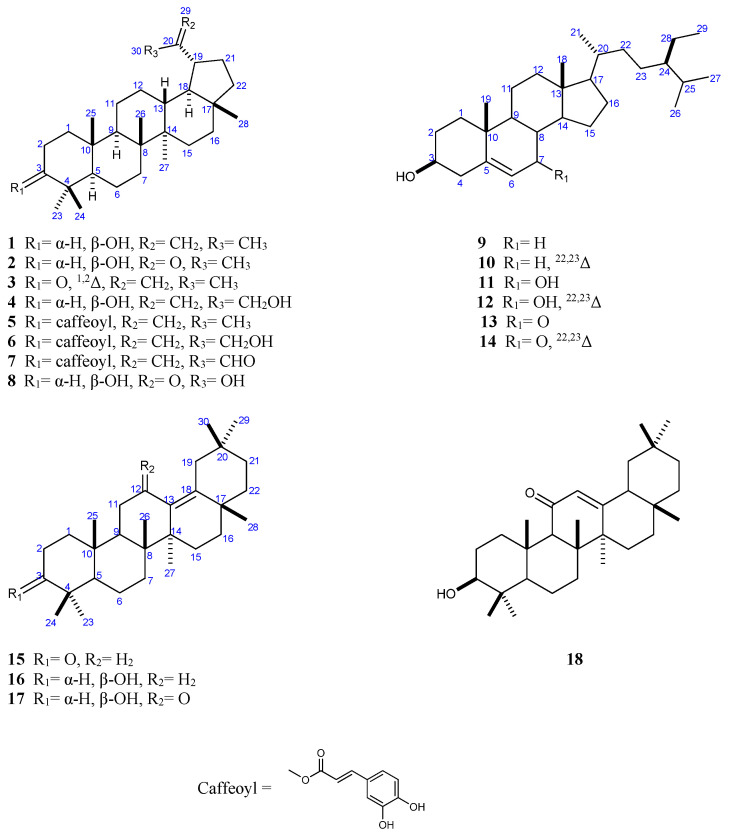
Structures of compounds (**1–18**) isolated from the aerial parts of *Dobera glabra*.

**Figure 2 plants-10-00119-f002:**
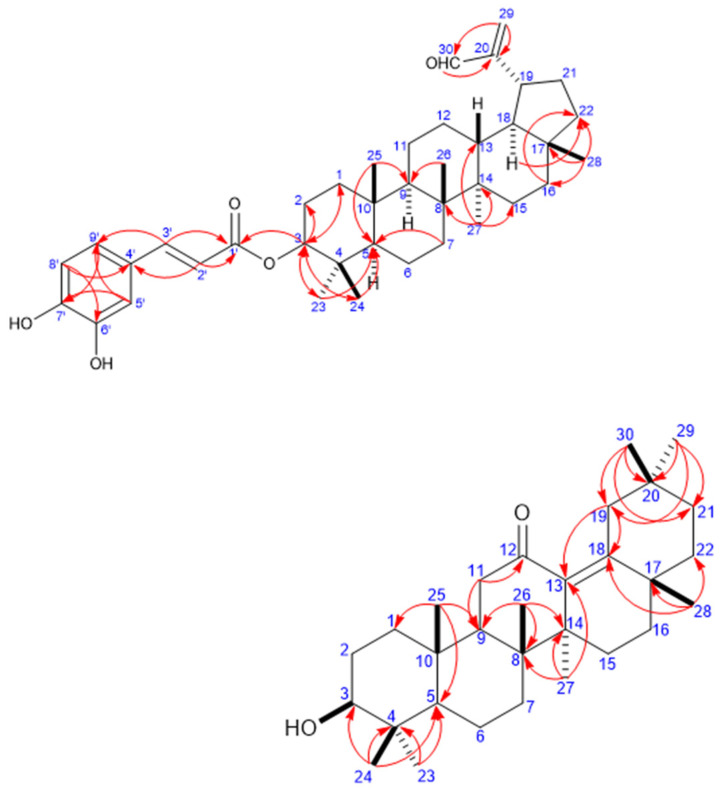
Key Heteronuclear multiple-bond correlation spectroscopy correlations (H → C) in compounds **7** and **11**.

**Figure 3 plants-10-00119-f003:**
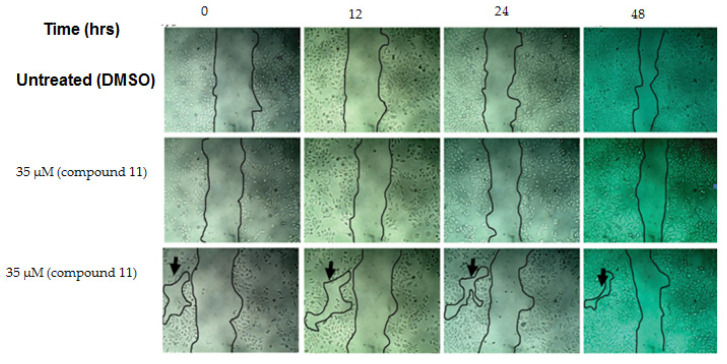
Effect of compound **11** on the wound healing of normal cells (*Vero* cell line). The cells were able to migrate in order to decrease the space between each other (black arrow), so that the wound size decreased after 48 h of incubation.

**Table 1 plants-10-00119-t001:** ^1^H (500 MHz) and ^13^C NMR (125 MHz) spectroscopic data of compounds **7** and **17** in CDCl_3_.

No.	7		17	
	*δ* _C_	*δ* _H_	*δ* _C_	*δ* _H_
1	38.4	1.78, m	39.5	1.62, m 1.01, m
2	23.9	1.80, m	27.8	1.61, m
3	81.5	4.72, t (8.1)	79.4	3.16, dd (11.0, 4.8)
4	38.1	-	39.9	-
5	55.4	0.92, m	56.7	0.82, m
6	18.3	1.65, m 1.53, m	19.4	1.67, m 1.52, m
7	34.3	1.52, m	35.0	1.52, m
8	40.9	-	42.0	-
9	50.2	1.36, m	51.6	1.74, m
10	37.1	-	38.4	-
11	21.0	1.47, m 1.30, m	41.5	2.30, m
12	27.7	1.14, m	209.9	-
13	37.8	1.07	140.6	-
14	42.8	-	46.6	-
15	27.4	1.82, m	26.2	1.89, m 1.12, m
16	35.4	1.65, m 1.57, m	36.9	1.40, m
17	43.4	-	36.2	-
18	50.2	1.36, m	149.6	-
19	37.8	1.77, m	40.5	1.67, d (12.7)
20	157.5	-	34.7	-
21	29.8	1.40, m	36.4	1.60, m 1.21, m
22	40.0	1.58, m 1.52, m	40.1	1.46, m
23	28.1	1.01, s	28.6	0.99, s
24	16.8	1.04, s	16.2	0.79, s
25	16.2	0.99, s	16.4	0.95, s
26	16.0	1.14, s	17.5	1.06, s
27	14.5	1.06, s	22.1	0.97, s
28	17.9	0.96, s	23.7	1.11, s
29	134.2	6.48, brs 6.11, brs	24.9	0.87, brs
30	196.0	9.65, s	32.7	0.93, s
1’	168.3	-		
2’	115.8	6.39, d (15.5)		
3’	145.2	7.70, d (15.5)		
4’	127.3	-		
5’	114.4	7.26, brs		
6’	144.4	-		
7’	146.9	-		
8’	115.5	7.02, d (7.5)		
9’	122.4	7.11, d (7.5)		

**Table 2 plants-10-00119-t002:** Cytotoxic activities of the different plant extract fractions and isolated compounds.

Fractions/Compounds	HCT-116 IC_50_ (µg/mL)	PC-3 IC_50_ (µg/mL)	HepG-2 IC_50_ (µg/mL)	VERO-B IC_50_ (µg/mL)
Hexane fraction	24 ± 0.34	24 ± 0.34	24 ± 0.34	82 ± 1.4
CHCl_3_ fraction	7.1 ± 0.23	7.1 ± 0.23	7.1 ± 0.23	80 ± 0.2
*n*-Butanol fraction	16 ± 1.31	16 ± 1.31	16 ± 1.31	78 ± 1.4
Crude ethanolic extract of stems	18 ± 0.02	18 ± 0.02	18 ± 0.02	86 ± 0.5
Crude ethanolic extract of leaves	19 ± 1.46	19 ± 1.46	19 ± 1.46	80 ± 1.4
**1**	>100	>100	>100	>100
**2**	>100	>100	>100	>100
**3**	>100	>100	>100	>100
**4**	>100	>100	>100	>100
**5**	59 ± 0.45	59 ± 0.45	59 ± 0.45	81 ± 1.1
**6**	68 ± 1.72	68 ± 1.72	68 ± 1.72	68 ± 0.3
**7**	62 ± 1.42	62 ± 1.42	62 ± 1.42	>100
**8**	>100	>100	>100	>100
**9**	>100	>100	>100	>100
**11**	8 ± 1.02	8 ± 1.02	8 ± 1.02	70 ± 0.5
**13**	22.3 ± 0.24	22.3 ± 0.24	22.3 ± 0.24	64 ± 1.1
Cisplatin	12.6 ± 2	5 ± 0.45	5 ± 1.5	11 ± 1.3

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
