# Peer review of "Sterols and Triterpenes from Dobera glabra Growing in Saudi Arabia and Their Cytotoxic Activity"

_plants, 2021, doi:10.3390/plants10010119_

Round 1
Reviewer 1 Report
It is opinion of the reviewer that this interesting and well prepared paper before acceptance needs several revisions. My individual comments are listed below.
23 – It should be “elucidated” instead of “determined”.
23 – The used spectroscopic methods should be mentioned.
200 – It should be “…mentioned enzyme at used condition”.
Table 2 – It should be “Crude ethanolic extract of stems”, “Crude ethanolic extract of leaves”.
241 – The model of the used a Bruker microTOF mass spectrometer?
243 – The used solvent for NMR?
244 – Thickness of the layer? Gel particle size?
245 – Mobile phase?
247 – C18 should added and characterized.
249 – Information about microplate reader should be completed.
How the column chromatography methods were monitored?
296 – “cis-” must be in italic.
321 – The volume of the sample?
322 – It should be “0.742 mU”.
References: The journal title and issue must be in italic; year in bold.
406 – “page number”?
Author Response
Reviewer 1
It is opinion of the reviewer that this interesting and well prepared paper before acceptance needs several revisions. My individual comments are listed below.
- Thank you for kind help.
23 – It should be “elucidated” instead of “determined”.
- Amended as:
Their structures were elucidated using a variety of spectroscopic techniques including 1D (1H, 13C, and DEPT-135 13C) and 2D (1H–1H COSY, 1H–13C HSQC, and 1H–13C HMBC) NMR and accurate mass spectroscopy
23 – The used spectroscopic methods should be mentioned.
- Amended as:
Their structures were elucidated using a variety of spectroscopic techniques including 1D (1H, 13C, and DEPT-13513C) and 2D (1H–1H COSY, 1H–13C HSQC, and 1H–13C HMBC) NMR and accurate mass spectroscopy
200 – It should be “…mentioned enzyme at used condition”.
- Amended
Table 2 – It should be “Crude ethanolic extract of stems”, “Crude ethanolic extract of leaves”.
- Amended
241 – The model of the used a Bruker microTOF mass spectrometer?
- Amended
accurate mass determination was achieved with a JEOL JMS-700 High-Resolution Mass Spectrophotometer (JEOL USA Inc., Peabody, MA, USA) with positive and negative mode
243 – The used solvent for NMR?
Done
244 – Thickness of the layer? Gel particle size?
A layer Thickness of 250 mm and a mean particle size 10-12 mm, with different sizes.
245 – Mobile phase?
n-hexane: EtOAc, CHCl3:EtOAc and CHCl3: MeOH for normal phase TLC while H2O: MeOH (10:90) was used for RP-18 TLC
247 – C18 should added and characterized.
LiChrorep RP-18 (25-40mm) for liquid chromatography.
249 – Information about microplate reader should be completed.
Microplate reader, Coming Inc., NY, USA, ELISA BioTek Lx800 microplate added to the text lines 256
How the column chromatography methods were monitored?
The columns were monitored by examination under uv lamp 254/366 nm followed by spraying by 15% H2SO4 in ethanol followed by heating at 120 0C
296 – “cis-” must be in italic.
Done
321 – The volume of the sample?
(100 µL/well): Done in the text
322 – It should be “0.742 mU”.
Done
References: The journal title and issue must be in italic; year in bold.
Done
406 – “page number”?
Page 237
Thank you very much for kind help and effort

Reviewer 2 Report
Comments and remarks:
- the title, please change it, please add the instrumental/analytical techniques used,
- in the introduction part should be highlighted the main aim of the paper, and additionally, what is the novelty of carried research work,
- in the Reference list there is lack of the most important papers regarding biological activity, chromatographic methods, instrumental techniques, biochemical methods as well,
- how do the Authors select the analytes? The rational of the choice of the selected biologically active compounds studied is missing and should be clearly discussed. Additionally, these analytes are not listed in the abstract section,
- what about validation of the obtained models (training and test sets)?,
- what is a value of mean square errors (MSE)?,
- figure 3 – please add the name of cells,
- chapters 3.7 and 3.8 should be more extended,
- there is lack of mass spectrometry description used,
- there is lack MS chromatograms/spectra, which will prove the obtain results.
Author Response
Reviewer 2
It is opinion of the reviewer that this interesting and well prepared paper before acceptance needs several revisions. My individual comments are listed below.
- Thank you for kind help.
- the title, please change it, please add the instrumental/analytical techniques used,
The title changed to “Sterols and triterpenes from Dobera glabra growing in Saudi Arabia
and their cytotoxic activity”
- in the introduction part should be highlighted the main aim of the paper, and additionally, what is the novelty of carried research work,
As part of our intensive search for new bioactive compounds in Saudi plants with a potential cytotoxic activity, in the current study we thoroughly investigated D. glabra grown in the wild in Saudi Arabia to identify its active constituents and assess their cytotoxic activity. The selected compounds were chosen for screening their biological activity based on their isolated amount and the reported cytotoxic activity of structurally-related steroids and triterpenes
- In the Reference list there is lack of the most important papers regarding biological activity, chromatographic methods, instrumental techniques, biochemical methods as well,
More references are added, such as:
literature reviewing of the titled plant revealed only one research paper dealing with the phytochemistry and biological activity of Dobera glabra and we have cited this paper in the list of references as follow
- Elkhateeb, A., Abdel Latif, R.R., Marzouk, M.M., Hussein, S.R., Kassem, M. E. S., Khalil, W. K. B., El-Ansari M.A. Flavonoid constituents of Dobera glabra leaves: amelioration impact against CCl4-induced changes in the genetic materials in male rats. Pharm. Biol., 2017, 55, 139–145.
In the field of natural product chemistry and biology there are common chromatographic techniques such as TLC, CC, CPTLC used for isolation and purification of secondary metabolites. For chromatographic methods and instrumental techniques including UV, IR, NMR and MS, please see references
- Muhammad I., Samoylenko V., Machumi F., Zakia M.A., Mohammed R., Hetta M.H., Gillum V. Preparation and application of reversed phase Chromatorotor for the isolation of natural products by centrifugal preparative chromatography. Prod. Commun. 2013, 8, 311–314.
- Kulkarni N., Mandhanya M., Jain D.K. Centrifugal thin layer chromatography. Asian j. pharm. life sci., 2011, 1, 294–300.
For biochemical methods
Cytotoxic activity
For Possible cytotoxicity mechanism of compound 11 please see references [34-37]
For Cell lines, cultures and tested compounds please see references [40-45]
For Phosphodiestrase inhibition investigation please see references [34-37]
For Wound healing assay please see reference [45]
- how do the Authors select the analytes? The rational of the choice of the selected biologically active compounds studied is missing and should be clearly discussed. Additionally, these analytes are not listed in the abstract section.
The analyte was selected as part of ongoing research project to explore the local Saudi plants aiming to find novel compounds with potential use as anticancer drug. The selected compounds were chosen for screening of biological activity based of their availability (isolated amount) and the reported cytotoxic activity of structurally -related steroids and triterpenes. Please kindly see the following references
Wen-Shu Wang, Kun Gao, Chun-Ming Wang and Zhong-Jian Jia. Cytotoxic triterpenes from Ligulariopsis shichuana Pharmazie 58 (2003) 2
Shi-Biao Wu, Qiu-Ying Bao, Wen-Xuan Wang, Yun Zhao, Gang Xia, Zheng Zhao, Huaqiang Zeng, Jin-Feng Hu Cytotoxic Triterpenoids and Steroids from the Bark of Melia azedarach Planta Med 2011; 77: 922–928.
Compounds (1-9, 11 and 13) were added to abstract
- what about validation of the obtained models (training and test sets)?,
Each experiment was repeated three times, and the standard deviation was calculated (±). The concentration that caused a 50% inhibition of the cell growth (IC50) was calculated for each compound or fraction. The growth of the cells was monitored and the images were acquired using Gx microscopes (GXMGXD202 Inverted Microscope) (10x Eyepiece) after staining with crystal violet
- What is a value of mean square errors (MSE)?,
In an analogy to standard deviation, taking the square root of MSE yields the root-mean-square error or root-mean-square deviation (RMSE or RMSD), which has the same units as the quantity being estimated; for an unbiased estimator, the RMSE is the square root of the variance, known as the standard error.
- figure 3 – please add the name of cells, (Vero cell line):
Done in the text
- chapters 3.7 and 3.8 should be more extended,
1HNMR and 13CNMR are listed in detail in Table 1
- there is lack of mass spectrometry description used,
Mass described in section 3.1
- there is lack MS chromatograms/spectra, which will prove the obtain results.
The MS chromatograms for the new isolated compounds were added to the supporting data.

Reviewer 3 Report
The article, from my point of view, is sufficiently novel and therefore can be published, since it characterizes two new triterpenes and isolates 10 other triterpenes already known from Dobera glabra. It uses spectroscopic techniques for structural characterization and finally investigates the cytotoxic activity against cisplatin of different plant extracts and isolated compounds, and suggests that compound 11 may be an effective anticancer agent candidate.
The introduction is well written and correctly states the objectives and the state of the art in relation to the active constituents of D. glabla.
Regarding the results and discussion, the structural elucidation of compounds 1 to 18 is highly valued, using 1D (1H, 74 13C, and DEPT-13513C) and 2D (1H – 1H COZY, 1H – 13C HSQC, and 1H - 75 13C HMBC) NMR (Figures 1S-46S) and especially of the new compounds 7 and 17. The figures and tables are correctly presented, for the new compounds 7 and 11.
In relation to cytotoxic activity, we believe that it is necessary to explain and discuss in more detail why compound 11 has a higher activity in relation to its structure, compared to the other compounds. Another important fact that should be discussed in more detail is the higher activity of all the extracts against the purest compounds and to verify if this conclusion has already been verified in the bibliography for extracts from other plants. It would be very useful to collect those articles and present them in the form of a new review table.
Author Response
Reviewer 3
It is opinion of the reviewer that this interesting and well prepared paper before acceptance needs several revisions. My individual comments are listed below.
- Thank you for kind help.
The article, from my point of view, is sufficiently novel and therefore can be published, since it characterizes two new triterpenes and isolates 10 other triterpenes already known from Dobera glabra. It uses spectroscopic techniques for structural characterization and finally investigates the cytotoxic activity against cisplatin of different plant extracts and isolated compounds, and suggests that compound 11 may be an effective anticancer agent candidate.
The introduction is well written and correctly states the objectives and the state of the art in relation to the active constituents of D. glabla.
Regarding the results and discussion, the structural elucidation of compounds 1 to 18 is highly valued, using 1D (1H, 13C, and DEPT-13513C) and 2D (1H – 1H COSY, 1H – 13C HSQC, and 1H - 13C HMBC) NMR (Figures 1S-46S) and especially of the new compounds 7 and 17. The figures and tables are correctly presented, for the new compounds 7 and 11.
In relation to cytotoxic activity, we believe that it is necessary to explain and discuss in more detail why compound 11 has a higher activity in relation to its structure, compared to the other compounds. Another important fact that should be discussed in more detail is the higher activity of all the extracts against the purest compounds and to verify if this conclusion has already been verified in the bibliography for extracts from other plants. It would be very useful to collect those articles and present them in the form of a new review table.
Response
The submitted study leads us to conduct a rigorous molecular biology-based study in conjunction with computer-assisted design to reveal and/or justify the activity of compound 11. Perhaps the higher activity of all extracts against pure compounds due to synergistic effect of all secondary metabolites in the whole extracts rather than one compound.
Please see the following reference
The last sentence adds to the manuscript lines 194-199.
Marco Mellado, Mauricio Soto, Alejandro Madrid, Iván Montenegro, Carlos Jara-Gutiérrez, Joan Villena, Enrique Werner, Patricio Godoy & Luis F. Aguilar In vitro antioxidant and antiproliferative effect of the extracts of Ephedra chilensis K Presl aerial parts. BMC Complementary and Alternative Medicine volume 19, Article number: 53 (2019)

Reviewer 4 Report
This paper reports the exploration of the bioactive compound and its activity of Dobera glabra from Saudi Arabia. The information was useful for further exploration of the plant. However, some improvements are suggested to the authors to increase the quality of the manuscript.
Title:
The title needs to be specified, it should explain the reported study. Maybe investigation of the new compounds and their activity.
Introduction:
This section needs to rewrite. Important explanation regarding the conducted study needs to be explained.
- In the first paragraph, an explanation related to Salvadora persica is not needed as it was not studied.
- The studied analyte has not been mentioned at all, even so the elaboration. The author should explain what kind of compound was studied and why it is important.
- The analysis was employed using some techniques and equipment. This should be well explained, why various techniques were used instead of choosing the best-suited one, what will the difference in technique and equipment tell us.
- The study also covered the biological activity of the studied compound, the justification on why the activity is chosen for the study should be included in the introduction.
- Finally, the author should present the purpose of the study in this section.
Result and Discussion:
- Line 183: The author mentioned 'the different fraction', if this was also the variables in the study, it also should be introduced in the introduction section, along with the justification.
- Line 185: What is the basis for delivering as good and moderate activity? Which parameter is used for the justification?
- Line 193: If the rate for the treated and the untreated cell were the same, then what is the excellence in using the proposed compound?
- Line 225: There has not been evaluation nor explanation regarding the side effect of the treatment, so this sentence is lack of supporting judgment.
- Figure 3: This figure is lacking information. This figure was mentioned in line 195 to be explaining the comparison of the treated and untreated cell, however, there is no information to which pictures shows the treated or the untreated cells. Note for the time difference should be inserted to make it clearer.
Material and Methods:
- Line 249: "Reagents, chemicals, and solvents" should be well explained.
- Point 3.2: Instead mentioned who conducts the specific activity, this section should describe the samples and explained the sample preparation, as in the following section the mentioned sample was in the dried and powdered form.
Conclusion:
- It was confusing as in the abstract and conclusion the author mentioned the finding of new compounds, however, these compounds were not further studied nor evaluated.
- Line 365: There has not been an evaluation nor explanation regarding the side effect of the treatment in the result and discussion section. Therefore, this is inaccurate to be mentioned in the conclusion.
Author Response
Reviewer 4
It is opinion of the reviewer that this interesting and well prepared paper before acceptance needs several revisions. My individual comments are listed below.
- Thank you for kind help.
This paper reports the exploration of the bioactive compound and its activity of Dobera glabra from Saudi Arabia. The information was useful for further exploration of the plant. However, some improvements are suggested to the authors to increase the quality of the manuscript.
Title:
The title needs to be specified; it should explain the reported study. Maybe investigation of the new compounds and their activity.
The title changed to “Sterols and triterpenes from Dobera glabra growing in Saudi Arabia
and their cytotoxic activity”
Introduction:
This section needs to rewrite. Important explanation regarding the conducted study needs to be explained.
More information was adding to introduction. As part of our intensive search for new bioactive compounds in Saudi plants, in the current study we thoroughly investigated D. glabra grown in the wild in Saudi Arabia to identify its active constituents and assess their cytotoxic activity. The selected compounds were chosen for screening of biological activity based on their availability and the reported cytotoxic activity of structurally -related steroids and triterpenes
- In the first paragraph, an explanation related to Salvadora persica is not needed as it was not studied.
The paragraph on Salvadora persica was deleted
- The studied analyte has not been mentioned at all, even so the elaboration. The author should explain what kind of compound was studied and why it is important.
It is mentioned in the abstract line 26
The compounds studied were steroids and triterpenes based on the reported similar activity for structurally-related compounds please kindly see the following references
Wen-Shu Wang, Kun Gao, Chun-Ming Wang and Zhong-Jian Jia. Cytotoxic triterpenes from Ligulariopsis shichuana Pharmazie 58 (2003) 2
Shi-Biao Wu, Qiu-Ying Bao, Wen-Xuan Wang, Yun Zhao, Gang Xia, Zheng Zhao, Huaqiang Zeng, Jin-Feng Hu Cytotoxic Triterpenoids and Steroids from the Bark of Melia azedarach Planta Med 2011; 77: 922–928.
Several paragraphs add to the introduction part
- The analysis was employed using some techniques and equipment. This should be well explained, why various techniques were used instead of choosing the best-suited one, what will the difference in technique and equipment tell us.
As you know the structural elucidation of natural products is achieved by different spectroscopic techniques including UV, IR MS and NMR. We have no choice for choosing one technique and exclude the others since all these techniques help us in the final confirmation of the structure ie. each technique give information differ from the other one and all cooperate at the end to confirm the exact final structure of the isolated compounds.
- The study also covered the biological activity of the studied compound, the justification on why the activity is chosen for the study should be included in the introduction.
- Finally, the author should present the purpose of the study in this section.
As you know Cancer is a major public health problem worldwide and is the second leading cause
of death in the United States. In our ongoing research projects, we are searching for new drugs entities able to use for treatment of contemporary disease including cancer, so that finding new drug lead is one our research goal that is why we screened the total extract and fraction as well as some of purely-available isolated compounds from the titled plants for their promising cytotoxic activity against different cell lines.
The following sentences was added to the introduction
Since cancer is a very dangerous diseases and one of the leading cause of death worldwide and
Result and Discussion:
- Line 183: The author mentioned 'the different fraction', if this was also the variables in the study, it also should be introduced in the introduction section, along with the justification.
Biological-guided fractionation of the different active fractions and the promising fractions were subjected to further chromatographic isolation and separation
The following sentences was added to the introduction part for justification
All obtained fractions were screened as well as some of the isolated compounds, chosen based on their availability and the reported activity of structurally -related steroids and triterpenes, for cytotoxic activity
- Line 185: What is the basis for delivering as good and moderate activity? Which parameter is used for the justification?
The IC50 of the positive control (cis-platin which is commonly used as standard)
- Line 193: If the rate for the treated and the untreated cell were the same, then what is the excellence in using the proposed compound?
The same rate on normal cell line. These data indicated that 11, the most active compound, is not cytotoxic to normal cells, and it is worth investigating further for use as a safe anticancer agent
- Line 225: There has not been evaluation nor explanation regarding the side effect of the treatment, so this sentence is lack of supporting judgment.
This is an interesting new trend in chemotherapy (Chemosenistization, https://www.researchgate.net/publication/301692043_Gingerols_and_cucurbitacins_as_new_chemosensitizers_for_resistant_breast_and_ovarian_cancer) whereas adding a drug with high safety margin will lead to decrease the dose of chemotherapy and subsequently reduce the side effects and enhance the activity.
- Figure 3: This figure is lacking information. This figure was mentioned in line 195 to be explaining the comparison of the treated and untreated cell, however, there is no information to which pictures shows the treated or the untreated cells. Note for the time difference should be inserted to make it clearer.
Material and Methods:
- Line 249: "Reagents, chemicals, and solvents" should be well explained.
To avoid the high percentage of similarity (pilgarism) we don’t mention details for reagents, chemical and solvents and we mentioned below;
Reagents for TLC was Anisaldehyde-sulfuric acid spray reagent
p-anisaldehyde (0.5 mL) was mixed with 10 mL of glacial acetic acid, followed by 85 mL of methanol and 5 mL of concentrated sulfuric acid in this order. The reagent was sprayed on TLC plates, which were then heated in oven at 105 °C until color development was complete.
Liebermann-Burchard for detection of steroids and /or triterpene
A portion of the compound was dissolved in 0.5 mL of hot acetic anhydride, cooled and under layered with one mL of concentrated sulfuric acid. An intense reddish-violet color, changed to dark green, was produced at the junction of the two layers.
The solvents used were n-hexane, chloroform, ethyl acetate and ethanol. General purpose reagents (GPR) were used for extraction processes, while analytical grade solvents were used for chromatographic separation and purification, purchased from Sigma, Aldrich, USA. Methanol HPLC grade was obtained from Fisher Scientific, UK. For HPTLC, AR grade ethyl acetate and ethanol, were procured from BDH, UK. Deuterated solvents such as CD3OD, CDCl3 and DMSO-d6, (99.8% atom deuterium), used NMR measurements, were obtained from Sigma, Aldrich, USA.
Solvents n- hexane: EtOAc, CHCl3:EtOAc and CHCl3: MeOH for normal phase TLC and column chromatography while H2O: MeOH (10:90) was used for Rp-18 TLC and MPLC column
- Point 3.2: Instead mentioned who conducts the specific activity, this section should describe the samples and explained the sample preparation, as in the following section the mentioned sample was in the dried and powdered form.
As you know, in the field of natural product chemistry and biology the used samples for study, in our case is Dobera glabra plant, should be identified and authenticate by well -recognized taxonomist. The name of the taxonomist should be mentioned in this part, then the plant material dried in shade until constant weight, grinded using toothed mill and then extracted by ethanol to give the total alcoholic extract.
Conclusion:
- It was confusing as in the abstract and conclusion the author mentioned the finding of new compounds, however, these compounds were not further studied nor evaluated.
Compounds 7 and 17 are new compounds both were extensively studied for identification of their new structure using different spectroscopic techniques including UV, IR, HRMS and NMR. Compound 7 tested for its cytotoxic activity against four cancerous cell line cells and does not show a promising cytotoxic activity comparing to the standard Cisplatin while compound 17 is not tested due to its lower yield and we are planning to isolate it again in the future to check its activity together with untested compounds.
- Line 365: There has not been an evaluation nor explanation regarding the side effect of the treatment in the result and discussion section. Therefore, this is inaccurate to be mentioned in the conclusion.
We delete this sentence from the conclusion
In case of using this compound as cytotoxic drugs it should thoroughly investigated towards its efficacy and of course side effect, bioavailability, suitable pharmaceutical dosage form, its derivatives…etc. to get final information towards this drug. So that this just a preliminary suggestion and need a further investigation as shown in the last paragraph of the conclusion.
Currently, a more rigorous in vivo study is underway, which is directed at obtaining more preclinical information, such as oral stability, bioavailability, and pharmacokinetic with anticipation of better activity and high safety margin.

Reviewer 5 Report
The authors submitted an original article dealing with phytochemical analysis of plant Dobera glabra. Additionally, selected plant extracts and isolated constituents were investigated for their possible cytotoxic activity against some cancer cell lines.
Strengths of the manuscript: The manuscript show novel information about this plant species constituent and their possible bioactivities. The authors use adequate and standard methods. Furthermore, the manuscript was prepared with care. Moreover, the data supports the conclusions. The manuscript represents a good and consistent phytochemical and pharmacognostic research approach.
Weaknesses: I found no serious errors or mistakes.
I recommend the manuscript for the publication in the journal Plants.
Author Response
Dear doctor
Thank you very much for your kind and sincere effort for reviewing our manuscript
Round 2
Reviewer 2 Report
-
Author Response
Dear doctor
Thank you very much for your kind effort and sincere help
Reviewer 4 Report
The author has made some revisions based on the comments, however, some unrevised ones still need to be improved.
- It is true that each technique served a different purpose. However, the justification regarding the chosen technique used in this study should be addressed to show that the technique is fit to the purpose of the study.
- Line 183: The additional information the author mentioned, has yet can not be found in the revised version.
- Point 3.2: to recognize taxonomist is good, but more detailed information on sample preparation and handling are more useful for the reader.
- The purpose of defining complete info of reagents, chemicals, and solvents is to be sure of the quality of the material. I believe the editorial boards would not define it as an act of plagiarism and wisely measure the similarly.
Author Response
The author has made some revisions based on the comments, however, some unrevised ones still need to be improved.
- It is true that each technique served a different purpose. However, the justification regarding the chosen technique used in this study should be addressed to show that the technique is fit to the purpose of the study.
In this study we used several techniques aiming to achieve our research objective goal. The techniques used for structure elucidation of the isolated compounds were:-
Infra-Red (IR) which is a very efficient technique for identification of the functional groups of the compounds such as NH, OH, aromatic ring, olefenic double bond, acetyleneic functionalities and functional group such as carbonyl either free or conjugated.
Additionally, we used the NMR spectroscopy, which is a highly useful technique for structural elucidation and directly tracing out the carbon skeleton of a natural product, by utilizing both one dimensional (1D NMR) and two dimensional (2D NMR). In the NMR we can identify the number of proton signals, carbon signals and the environments nearby each one which will be affected by the chemical environment of the substituents. Two dimensional NMR also another advanced NMR technique which locate each proton to its corresponding carbon, determine the coupled protons as well as giving two and three bond correlations between each proton and the nearby carbon. Finally, the spectroscopic data obtained from IR, UV, NMR are employed together with high resolution HRMS which provide the molecular formula and molecular weight of the compound in 4 to 5 decimals.
.
- Line 183: The additional information the author mentioned, has yet cannot be found in the revised version.
Add to the introduction lines 71-74 and line 193
- Point 3.2: to recognize taxonomist is good, but more detailed information on sample preparation and handling are more useful for the reader.
The following sentence add to lines 279-281
The undesirable parts of plant material were removed. Aerial parts were dried in air- shade till constant weight and grinded using toothed mill followed by sifting using suitable mesh to give a homogenous particle size powder for subsequent efficient extraction.
- The purpose of defining complete info of reagents, chemicals, and solvents is to be sure of the quality of the material. I believe the editorial boards would not define it as an act of plagiarism and wisely measure the similarly.
